# Hybrid Macro/Micro Level Backpropagation for Training Deep Spiking Neural Networks

**Yingyezhe Jin**
Texas A&M University
College Station, TX 77843
jyyz@tamu.edu

**Wenrui Zhang**
Texas A&M University
College Station, TX 77843
zhangwenrui@tamu.edu

**Peng Li**
Texas A&M University
College Station, TX 77843
pli@tamu.edu

## Abstract

Spiking neural networks (SNNs) are positioned to enable spatio-temporal information processing and ultra-low power event-driven neuromorphic hardware. However, SNNs are yet to reach the same performances of conventional deep artificial neural networks (ANNs), a long-standing challenge due to complex dynamics and non-differentiable spike events encountered in training. The existing SNN error backpropagation (BP) methods are limited in terms of scalability, lack of proper handling of spiking discontinuities, and/or mismatch between the rate-coded loss function and computed gradient. We present a hybrid macro/micro level backpropagation (HM2-BP) algorithm for training multi-layer SNNs. The temporal effects are precisely captured by the proposed *spike-train level post-synaptic potential* (S-PSP) at the microscopic level. The rate-coded errors are defined at the macroscopic level, computed and back-propagated across both macroscopic and microscopic levels. Different from existing BP methods, HM2-BP directly computes the gradient of the rate-coded loss function w.r.t tunable parameters. We evaluate the proposed HM2-BP algorithm by training deep fully connected and convolutional SNNs based on the static MNIST [14] and dynamic neuromorphic N-MNIST [26]. HM2-BP achieves an accuracy level of $99.49\%$ and $98.88\%$ for MNIST and N-MNIST, respectively, outperforming the best reported performances obtained from the existing SNN BP algorithms. Furthermore, the HM2-BP produces the highest accuracies based on SNNs for the EMNIST [3] dataset, and leads to high recognition accuracy for the 16-speaker spoken English letters of TI46 Corpus [16], a challenging spatio-temporal speech recognition benchmark for which no prior success based on SNNs was reported. It also achieves competitive performances surpassing those of conventional deep learning models when dealing with asynchronous spiking streams.

## 1   Introduction

In spite of recent success in deep neural networks (DNNs) [5, 9, 13], it is believed that biological brains operate rather differently. Compared with DNNs that lack processing of spike timing and event-driven operations, biologically realistic spiking neural networks (SNNs) [11, 19] provide a promising paradigm for exploiting spatio-temporal patterns for added computing power, and enable ultra-low power event-driven neuromorphic hardware [1, 7, 20]. There are theoretical evidences supporting that SNNs possess greater computational power over traditional artificial neural networks (ANNs) [19]. SNNs are yet to achieve a performance level on a par with deep ANNs for practical applications. The error backpropagation [28] is very successful in training ANNs. Attaining the same success of backpropagation (BP) for SNNs is challenged by two fundamental issues: complex temporal dynamics and non-differentiability of discrete spike events.

**Problem Formulation:** As a common practice in SNNs, the rate coding is often adopted to define a loss for each training example at the output layer [15, 32]

$$E = \frac{1}{2}||\mathbf{o} - \mathbf{y}||_2^2, \tag{1}$$

where $\mathbf{o}$ and $\mathbf{y}$ are vectors specifying the actual and desired (label) firing counts of the output neurons. Firing counts are determined by the underlying firing events, which are adjusted discretely by tunable weights, resulting in great challenges in computing the gradient of the loss with respect to the weights.

**Prior Works:** There exist approaches that stay away from the SNN training challenges by first training an ANN and then approximately converting it to an SNN [6, 7, 10, 24]. [25] takes a similar approach which treats spiking neurons almost like non-spiking ReLU units. The accuracy of those methods may be severely compromised because of imprecise representation of timing statistics of spike trains. Although the latest ANN-to-SNN conversion approach [27] shows promise, the problem of direct training of SNNs remains unsolved.

The SpikeProp algorithm [2] is the first attempt to train an SNN by operating on discontinuous spike activities. It specifically targets temporal learning for which derivatives of the loss w.r.t. weights are explicitly derived. However, SpikeProp is very much limited to single-spike learning, and its successful applications to realistic benchmarks have not been demonstrated. Similarly, [33] proposed a temporal training rule for understanding learning in SNNs. More recently, the backpropagation approaches of [15] and [32] have shown competitive performances. Nevertheless, [15] lacks explicit consideration of temporal correlations of neural activities. Furthermore, it does not handle discontinuities occurring at spiking moments by treating them as noise while only computing the error gradient for the remaining smoothed membrane voltage waveforms instead of the rate-coded loss. [32] addresses the first limitation of [15] by performing BPTT [31] to capture temporal effects. However, similar to [15], the error gradient is computed for the continuous membrane voltage waveforms resulted from smoothing out all spikes, leading to inconsistency w.r.t the rate-coded loss function. In summary, the existing SNNs BP algorithms have three major limitations: i) suffering from limited learning scalability [2], ii) either staying away from spiking discontinuities (e.g. by treating spiking moments as noise [15]) or deriving the error gradient based on the smoothed membrane waveforms [15, 32], and therefore iii) creating a mismatch between the computed gradient and targeted rate-coded loss [15, 32].

**Paper Contributions:** We derive the gradient of the rate-coded error defined in (1) by decomposing each derivative into two components

$$\frac{\partial E}{\partial w_{ij}} = \underbrace{\frac{\partial E}{\partial a_i}}_{\text{bp over firing rates}} \times \underbrace{\frac{\partial a_i}{\partial w_{ij}}}_{\text{bp over spike trains}} , \tag{2}$$

where $a_i$ is the (weighted) aggregated membrane potential for the post-synaptic neuron $i$ per (11). As such, we propose a novel hybrid macro-micro level backpropagation (HM2-BP) algorithm which performs error backpropagation across two levels: 1) backpropagation over firing rates (**macro-level**), 2) backpropagation over spike trains (**micro-level**), and 3) backpropagation based on interactions between the two levels, as illustrated in Fig. 1.

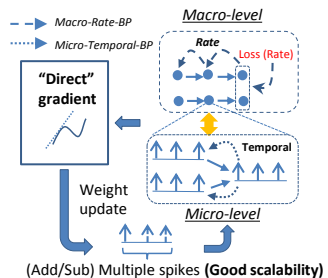

Figure 1: Hybrid macro-micro level backpropagation.

At the microscopic level, for each pre/post-synaptic spike train pair, we precisely compute the *spike-train level post-synaptic potential*, referred to as *S-PSP* throughout this paper, to account for the temporal contribution of the given pre-synaptic spike train to the firings of the post-synaptic neuron based on exact spike times. At the macroscopic level, we back-propagate the errors of the defined rate-based loss by aggregating the effects of spike trains on each neuron's firing count via the use of S-PSPs, and leverage this as a practical way of linking spiking events to firing rates. To assist backpropagation, we further propose a decoupled model of the S-PSP for disentangling the effects of firing rates and spike-train timings to allow differentiation of the S-PSP w.r.t. pre and post-synaptic firing rates at the micro-level. As a result, our HM2-BP approach is able to evaluate the direct impact of weight changes on the rate-coded loss function. Moreover, the resulting weight updates in each training iteration can lead to introduction or disappearance of multiple spikes.

We evaluate the proposed BP algorithm by training deep fully connected and convolutional SNNs based on the static MNIST [14], dynamic neuromorphic N-MNIST [26], and EMNIST [3] datasets. Our BP algorithm achieves an accuracy level of 99.49%, 98.88% and 85.57% for MNIST, N-MNIST and EMNIST, respectively, outperforming the best reported performances obtained from the existing

SNN BP algorithms. Furthermore, our algorithm achieves high recognition accuracy of 90.98% for the 16-speaker spoken English letters of TI46 Corpus [16], a challenging spatio-temporal speech recognition benchmark for which no prior success based on SNNs was reported.

## 2  Hybrid Macro-Micro Backpropagation

The complex dynamics generated by spiking neurons and non-differentiable spike impulses are two fundamental bottlenecks for training SNNs using backpropagation. We address these difficulties at both macro and micro levels.

### 2.1  Micro-level Computation of Spiking Temporal Effects

The leaky integrate-and-fire (LIF) model is one of the most prevalent choices for describing dynamics of spiking neurons, where the neuronal membrane voltage $u_i(t)$ at time $t$ for the neuron $i$ is given by

$$\tau_m \frac{u_i(t)}{dt} = -u_i(t) + R\,I_i(t), \tag{3}$$

where $I_i(t)$ is the input current, $R$ the effective leaky resistance, $C$ the effective membrane capacitance, and $\tau_m = RC$ the membrane time constant. A spike is generated when $u_i(t)$ reaches the threshold $\nu$. After that $u_i(t)$ is reset to the resting potential $u_r$, which equals to 0 in this paper. Each post-synaptic neuron $i$ is driven by a post-synaptic current of the following general form

$$I_i(t) = \sum_j w_{ij} \sum_f \alpha(t - t_j^{(f)}), \tag{4}$$

where $w_{ij}$ is the weight of the synapse from the pre-synaptic neuron $j$ to the neuron $i$, $t_j^{(f)}$ denotes a particular firing time of the neuron $j$. We adopt a first order synaptic model with time constant $\tau_s$

$$\alpha(t) = \frac{q}{\tau_s} \exp\left(-\frac{t}{\tau_s}\right) H(t), \tag{5}$$

where $H(t)$ is the Heaviside step function, and $q$ the total charge injected into the post-synaptic neuron $i$ through a synapse of a weight of 1. Let $\hat{t}_i$ denote the last firing time of the neuron $i$ w.r.t time $t$: $\hat{t}_i = \hat{t}_i(t) = \max\{t_i | t_i^{(f)} < t\}$. Plugging (4) into (3) and integrating (3) with $u(\hat{t}_i) = 0$ as its initial condition, we map the LIF model to the Spike Response Model (SRM) [8]

$$u_i(t) = \sum_j w_{ij} \sum_f \epsilon\left(t - \hat{t}_i, t - t_j^{(f)}\right), \tag{6}$$

with

$$\epsilon(s, t) = \frac{1}{C} \int_0^s \exp\left(-\frac{t'}{\tau_m}\right) \alpha\left(t - t'\right) \mathrm{d}t'. \tag{7}$$

Since $q$ and $C$ can be absorbed into the synaptic weights, we set $q = C = 1$. Integrating (7) yields

$$\epsilon(s, t) = \frac{\exp(-\max(t - s, 0)/\tau_s)}{1 - \frac{\tau_s}{\tau_m}} \left[\exp\left(-\frac{\min(s, t)}{\tau_m}\right) - \exp\left(-\frac{\min(s, t)}{\tau_s}\right)\right] H(s)H(t). \tag{8}$$

$\epsilon$ is interpreted as the normalized (by synaptic weight) *post-synaptic potential*, which is evoked by a single firing spike of the pre-synaptic neuron $j$.

For any time $t$, the exact "contribution" of the neuron $j$'s spike train to the neuron $i$'s post-synaptic potential is given by summing (8) over all pre-synaptic spike times $t_j^{(f)}$, $t_j^{(f)} < t$. We particularly concern the contribution right before each post-synaptic firing time $t_i^{(f)}$ when $u_i(t_i^{(f)}) = \nu$, which we denote by $e_{i|j}(t_i^{(f)})$. Summing $e_{i|j}(t_i^{(f)})$ over all post-synaptic firing times gives the *total* contribution of the neuron $j$'s spike-train to the firing activities of the neuron $i$ as shown in Fig. 2

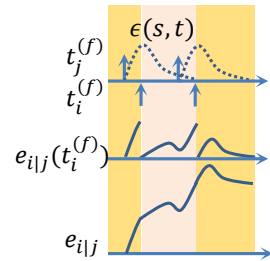

Figure 2: The computation of the S-PSP.

$$e_{i|j} = \sum_{t_i^{(f)}} \sum_{t_j^{(f)}} \epsilon(t_i^{(f)} - \hat{t}_i^{(f)}, t_i^{(f)} - t_j^{(f)}), \tag{9}$$

where $\hat{t}_i^{(f)} = \hat{t}_i^{(f)}(t_i^{(f)})$ denotes the last post-synaptic firing time before $t_i^{(f)}$.

Importantly, we refer to $e_{i|j}$ as the (normalized) *spike-train level post-synaptic potential* (S-PSP). As its name suggests, S-PSP characterizes the aggregated influence of the pre-synaptic neuron on the post-synaptic neuron's firings at the level of spike trains, providing a basis for relating firing counts to spike events and enabling scalable SNN training that adjusts spike trains rather than single spikes. Clearly, each S-PSP $e_{i|j}$ depends on both rate and temporal information of the pre/post spike trains. To assist the derivation of our BP algorithm, we make the dependency of $e_{i|j}$ on the pre/post-synaptic firing counts $o_i$ and $o_j$ explicit although $o_i$ and $o_j$ are already embedded in the spike trains

$$e_{i|j} = f(o_j, o_i, \mathbf{t}_j^{(f)}, \mathbf{t}_i^{(f)}), \tag{10}$$

where $\mathbf{t}_j^{(f)}$ and $\mathbf{t}_i^{(f)}$ represent the pre and post-synaptic timings, respectively. Summing the weighted S-PSPs from all pre-synaptic neurons results in the *total post-synaptic potential* (T-PSP) $a_i$, which is directly correlated to the neuron $i$'s firing count

$$a_i = \sum_j w_{ij} \, e_{i|j}. \tag{11}$$

## 2.2 Error Backpropagation at Macro and Micro Levels

It is evident that the total post-synaptic potential $a_i$ must be no less than the threshold $\nu$ in order to make the neuron $i$ fire at least once, and the total firing count is $\lfloor \frac{a_i}{\nu} \rfloor$. We relate the firing count $o_i$ of the neuron $i$ to $a_i$ approximately by

$$o_i = g(a_i) = \left\lfloor \frac{a_i}{\nu} \right\rfloor = \left\lfloor \frac{\sum_j w_{ij} \, e_{i|j}}{\nu} \right\rfloor \approx \frac{\sum_j w_{ij} \, e_{i|j}}{\nu}, \tag{12}$$

where the rounding error would be insignificant when $\nu$ is small. Despite that (12) is linear in S-PSPs, it is the interaction between the S-PSPs through nonlinearities hidden in the micro-level LIF model that leads to a given firing count $o_i$. Missing from the existing works [15, 32], (12) serves as an important bridge connecting the aggregated micro-level temporal effects with the macro-level count of discrete firing events. In a vague sense, $a_i$ and $o_i$ are analogies to pre-activation and activation in the traditional ANNs, respectively, although they are not directly comparable. (12) allows for rate-coded error backpropagation on top of discrete spikes across the macro and micro levels.

Using (12), the macro-level rate-coded loss of (1) is rewritten as

$$E = \frac{1}{2}||\mathbf{o} - \mathbf{y}||_2^2 = \frac{1}{2}||g(\mathbf{a}) - \mathbf{y}||_2^2, \tag{13}$$

where $\mathbf{y}$, $\mathbf{o}$ and $\mathbf{a}$ are vectors specifying the desired firing counts (label vector), the actual firing counts, and the weighted sums of S-PSP of the output neurons, respectively. We now derive the gradient of $E$ w.r.t $w_{ij}$ at each layer of an SNN.

- **Output layer:** For the $i_{th}$ neuron in the output layer $m$, we have

$$\frac{\partial E}{\partial w_{ij}} = \underbrace{\frac{\partial E}{\partial a_i^m}}_{\text{macro-level bp}} \times \underbrace{\frac{\partial a_i^m}{\partial w_{ij}}}_{\text{micro-level bp}}, \tag{14}$$

where variables associated with neurons in the layer $m$ have $m$ as the superscript. As shown in Fig. 3, the first term of (14) represents the macro-level backpropagation of the rate-coded error with the second term being the micro-level error backpropagation. From (13), the macro-level error backpropagation is given by

$$\delta_i^m = \frac{\partial E}{\partial a_i^m} = (o_i^m - y_i^m) \, g'(a_i^m) = \frac{o_i^m - y_i^m}{\nu}. \tag{15}$$

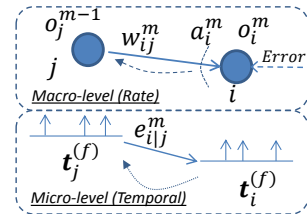

Figure 3: Macro/micro backpropagation in the output layer.

Similar to the conventional backpropagation, we use $\delta_i^m$ to denote the back propagated error. According to (11) and (10), $a_i^m$ can be unwrapped as

$$a_i^m = \sum_{j=1}^{r^{m-1}} w_{ij} \, e_{i|j}^m = \sum_{j=1}^{r^{m-1}} w_{ij} \, f(o_j^{m-1}, o_i^m, \mathbf{t}_j^{(f)}, \mathbf{t}_i^{(f)}), \tag{16}$$

where $r^{m-1}$ is the number of neurons in the $(m-1)_{th}$ layer. Differentiating (16) and making use of (12) leads to the micro-level error propagation based on the total post-synaptic potential (T-PSP) $a_i^m$

$$\frac{\partial a_i^m}{\partial w_{ij}} = \frac{\partial}{\partial w_{ij}} \left( \sum_{j=1}^{r^{m-1}} w_{ij} \, e_{i|j}^m \right) = e_{i|j}^m + \sum_{l=1}^{r^{m-1}} w_{il} \frac{\partial e_{i|l}^m}{\partial o_i^m} \frac{\partial o_i^m}{\partial w_{ij}} = e_{i|j}^m + \frac{e_{i|j}^m}{\nu} \sum_{l=1}^{r^{m-1}} w_{il} \frac{\partial e_{i|l}^m}{\partial o_i^m}. \quad (17)$$

Although the network is feed-forward, there are non-linear interactions between S-PSPs. The second term of (17) captures the hidden dependency of the S-PSPs on the post-synaptic firing count $o_i^m$.

• **Hidden layers:** For the $i_{th}$ neuron in the hidden layer $k$, we have

$$\frac{\partial E}{\partial w_{ij}} = \underbrace{\frac{\partial E}{\partial a_i^k}}_{\text{macro-level bp}} \times \underbrace{\frac{\partial a_i^k}{\partial w_{ij}}}_{\text{micro-level bp}} = \delta_i^k \, \frac{\partial a_i^k}{\partial w_{ij}}. \quad (18)$$

The macro-level error backpropagation at a hidden layer is much more involved as in Fig. 4

$$\delta_i^k = \frac{\partial E}{\partial a_i^k} = \sum_{l=1}^{r^{k+1}} \frac{\partial E}{\partial a_l^{k+1}} \frac{\partial a_l^{k+1}}{\partial a_i^k} = \sum_{l=1}^{r^{k+1}} \delta_l^{k+1} \frac{\partial a_l^{k+1}}{\partial a_i^k}. \quad (19)$$

According to (11) , (10) and (12), we unwrap $a_l^{k+1}$ and get

$$a_l^{k+1} = \sum_{p=1}^{r^k} w_{lp} \, e_{l|p}^{k+1} = \sum_{p=1}^{r^k} w_{lp} \, f(o_p^k, o_l^{k+1}, \mathbf{t}_p^{(f)}, \mathbf{t}_l^{(f)}) = \sum_{p=1}^{r^k} w_{lp} \, f(g(a_p^k), o_l^{k+1}, \mathbf{t}_p^{(f)}, \mathbf{t}_l^{(f)}). \quad (20)$$

Therefore, $\frac{\partial a_l^{k+1}}{\partial a_i^k}$ becomes

$$\frac{\partial a_l^{k+1}}{\partial a_i^k} = w_{li} \frac{\partial e_{l|i}^{k+1}}{\partial o_i^k} \frac{\partial o_i^k}{\partial a_i^k} = w_{li} \frac{\partial e_{l|i}^{k+1}}{\partial o_i^k} g'(a_i^k) = \frac{w_{li}}{\nu} \frac{\partial e_{l|i}^{k+1}}{\partial o_i^k}, \quad (21)$$

where the dependency of $e_{l|i}^{k+1}$ on the pre-synaptic firing count $o_i^k$ is considered but the one on the firing timings are ignored, which is supported by the decoupled S-PSP model in (25). Plugging (21) into (19), we have

$$\delta_i^k = \frac{1}{\nu} \sum_{l=1}^{r^{k+1}} \delta_l^{k+1} w_{li} \frac{\partial e_{l|i}^{k+1}}{\partial o_i^k}. \quad (22)$$

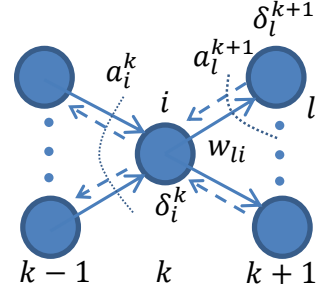

Figure 4: Macro-level backpropagation at a hidden layer.

The micro-stage backpropagation at hidden layers is identical to that at the output layer, i.e. (17). Finally, we obtain the derivative of $E$ with respect to $w_{ij}$ as follows

$$\frac{\partial E}{\partial w_{ij}} = \delta_i^k e_{i|j}^k \left( 1 + \frac{1}{\nu} \sum_{l=1}^{r^{k-1}} w_{il} \frac{\partial e_{i|l}^k}{\partial o_i^k} \right), \quad (23)$$

where

$$\delta_i^k = \begin{cases} \frac{o_i^m - y_i^m}{\nu} & \text{for output layer,} \\ \frac{1}{\nu} \sum_{l=1}^{r^{k+1}} \delta_l^{k+1} w_{li} \frac{\partial e_{l|i}^{k+1}}{\partial o_i^k} & \text{for hidden layers.} \end{cases} \quad (24)$$

Unlike [15, 32], here decomposing the rate-coded error backpropagation into the macro and micro levels enables computation of the gradient of the actual loss function with respect to the tunable weights, leading to highly competitive performances. Our HM2-BP algorithm can introduce/remove multiple spikes by one update, greatly improving learning efficiency in comparison with SpikeProp [2]. To complete the derivation of HM2-BP, derivatives in the forms of $\frac{\partial e_{i|j}^k}{\partial o_i^k}$ and $\frac{e_{i|j}^k}{\partial o_j^k}$ as needed in (17) and (22) are yet to be estimated, which is non-trivial as shall be presented in Section 2.3.

## 2.3 Decoupled Micro-Level Model for S-PSP

The derivatives of the S-PSP $e_{i|j}^k$ with respect to the pre and post-synaptic neuron firing counts are key components in our HM2-BP rule. According to (9), the S-PSP $e_{i|j}^k$ is dependent on both rate and temporal information of the pre and post-synaptic spikes. The firing counts of pre and post-synaptic neurons (i.e., the rate information) are represented by the two nested summations in (9). The exact firing timing information determines the (normalized) post-synaptic potential $\epsilon$ of each pre/post-synaptic spike train pair as seen from (8). The rate and temporal information of spike trains are strongly coupled together, making the exact computation of $\frac{\partial e_{i|j}^k}{\partial o_i^k}$ and $\frac{e_{i|j}^k}{\partial o_j^k}$ challenging.

To address this difficulty, we propose a decoupled model for $e_{i|j}^k$ to untangle the rate and temporal effects. The model is motivated by the observation that $e_{i|j}^k$ is linear in both $o_j^k$ and $o_i^k$ in the limit of high firing counts. For finite firing rates, we decompose $e_{i|j}^k$ into an asymptotic rate-dependent effect using the product of $o_j^k$ and $o_i^k$ and a correction factor $\hat{\alpha}$ accounting for temporal correlations between the pre and post-synaptic spike trains

$$e_{i|j}^k = \hat{\alpha}(\mathbf{t}_j^{(f)}, \mathbf{t}_i^{(f)}) o_j^k \, o_i^k. \tag{25}$$

$\hat{\alpha}$ is a function of exact spike timing. Since the SNN is trained incrementally with small weight updates set by a well-controlled learning rate, $\hat{\alpha}$ does not change substantially by one training iteration. Therefore, we approximate $\hat{\alpha}$ by using the values of $e_{i|j}^k$, $o_j^k$, and $o_i^k$ available before the next training update by

$$\hat{\alpha}(\mathbf{t}_j^{(f)}, \mathbf{t}_i^{(f)}) \approx \frac{e_{i|j}^k}{o_j^k o_i^k}.$$

With the micro-level temporal effect considered by $\hat{\alpha}$, we estimate the two derivatives by

$$\frac{\partial e_{i|j}^k}{\partial o_i^k} \approx \hat{\alpha} \, o_j^k = \frac{e_{i|j}^k}{o_i^k}, \quad \frac{\partial e_{i|j}^k}{\partial o_j^k} \approx \hat{\alpha} \, o_i^k = \frac{e_{i|j}^k}{o_j^k}.$$

Our hybrid training method follows the typical backpropagation methodology. First of all, a forward pass is performed by analytically simulating the LIF model (3) layer by layer. Then the firing counts of the output layer are compared with the desirable firing levels to compute the macro-level error. After that, the error in the output layer is propagated backwards at both the macro and micro levels to determine the gradient. Finally, an optimization method (e.g. Adam [12]) is used to update the network parameters given the computed gradient.

## 3 Experiments and Results

**Experimental Settings and Datasets** The weights of the experimented SNNs are randomly initialized by using the uniform distribution $U[-a, a]$, where $a$ is 1 for fully connected layers and $0.5$ for convolutional layers. We use fixed firing thresholds in the range of 5 to 20 depending on the layer. We adopt the exponential weight regularization scheme in [15] and introduce the lateral inhibition in the output layer to speed up training convergence [15], which slightly modifies the gradient computation for the output layer (see **Supplementary Material**). We use Adam [12] as the optimizer and its parameters are set according to the original Adam paper. We impose greater sample weights for incorrectly recognized data points during the training as a supplement to the Adam optimizer. More training settings are reported in the released source code.

The MNIST handwritten digit dataset [14] consists of 60k samples for training and 10k for testing, each of which is a $28 \times 28$ grayscale image. We convert each pixel value of a MNIST image into a spike train using Poisson sampling based on which the probability of spike generation is proportional to the pixel intensity. The N-MNIST dataset [26] is a neuromorphic version of the MNIST dataset generated by tilting a Dynamic Version Sensor (DVS) [17] in front of static digit images on a computer monitor. The movement induced pixel intensity changes at each location are encoded as spike trains. Since the intensity can either increase or decrease, two kinds of ON- and OFF-events spike events are recorded. Due to the relative shifts of each image, an image size of $34 \times 34$ is produced. Each sample of the N-MNIST is a spatio-temporal pattern with $34 \times 34 \times 2$ spike sequences lasting for

$300ms$. We reduce the time resolution of the N-MNIST samples by $600x$ to speed up simulation. The Extended MNIST-Balanced (EMNIST) [3] dataset, which includes both letters and digits, is more challenging than MNIST. EMNIST has 112,800 training and 18,800 testing samples for 47 classes. We convert and encode EMNIST in the same way as we do for MNIST. We also use the 16-speaker spoken English letters of TI46 Speech corpus [16] to benchmark our algorithm for demonstrating its capability of handling spatio-temporal patterns. There are 4,142 and 6,628 spoken English letters for training and testing, respectively. The continuous temporal speech waveforms are first preprocessed by Lyon's ear model [18] and then encoded into 78 spike trains using the BSA algorithm [29].

We train each network for 200 epochs except for ones used for EMNIST, where we use 50 training epochs. The best recognition rate of each setting is collected and each experiment is run for at least five times to report the error bar. For each setting, we also report the best performance over all the conducted experiments.

**Fully Connected SNNs for the Static MNIST**  Using Poisson sampling, we encode each $28 \times 28$ image of the MNIST dataset into a 2D $784 \times L$ binary matrix, where $L = 400ms$ is the duration of each spike sequence, and a $1$ in the matrix represents a spike. The simulation time step is set to be $1ms$. No pre-processing or data augmentation is done in our experiments. Table 1 compares the performance of SNNs trained by the proposed HM2-BP rule with other algorithms. HM2-BP achieves $98.93\%$ test accuracy, outperforming STBP [32], which is the best previously reported algorithm for fully-connected SNNs. The proposed rule also achieves the best accuracy earlier than STBP (100 epochs v.s. 200 epochs). We attribute the overall improvement to the hybrid macro-micro processing that handles the temporal effects and discontinuities at two levels in a way such that explicit back-propagation of the rate-coded error becomes possible and practical.

Table 1: Comparison of different SNN models on MNIST

| Model | Hidden layers | Accuracy | Best | Epochs |
|---|---|---|---|---|
| Spiking MLP (converted[*]) [24] | 500-500 | 94.09% | 94.09% | 50 |
| Spiking MLP (converted[*]) [10] | 500-200 | 98.37% | 98.37% | 160 |
| Spiking MLP (converted[*]) [6] | 1200-1200 | 98.64% | 98.64% | 50 |
| Spiking MLP [25] | 300-300 | 97.80% | 97.80% | 50 |
| Spiking MLP [15] | 800 | 98.71%[a] | 98.71% | 200 |
| Spiking MLP (STBP) [32] | 800 | 98.89% | 98.89% | 200 |
| Spiking MLP (this work) | 800 | **98.84 $\pm$ 0.02%** | **98.93%** | **100** |

We only compare SNNs without any pre-processing (i.e., data augmentation) except for [24].
[*] means the model is converted from an ANN. [a] [15] achieves $98.88\%$ with hidden layers of 300-300.

**Fully Connected SNNs for N-MNIST**  The simulation time step is $0.6ms$ for N-MNIST. Table 2 compares the results obtained by different models on N-MNIST. The first two results are obtained by the conventional CNNs with the frame-based method, which accumulates spike events over short time intervals as snapshots and recognizes digits based on sequences of snapshot images. The relative poor performances of the first two models may be attributed to the fact that the frame-based representations tend to be blurry and do not fully exploit spatio-temporal patterns of the input. The two non-spiking LSTM models, which are trained directly on spike inputs, do not perform too well, suggesting that LSTMs may be incapable of dealing with asynchronous and sparse spatio-temporal spikes. The SNN trained by our proposed approach naturally processes spatio-temporal spike patterns, achieving the start-of-the-art accuracy of $98.88\%$, outperforming the previous best ANN ($97.38\%$) and SNN ($98.78\%$) with significantly less training epochs required.

**Spiking Convolution Network for the Static MNIST**  We construct a spiking CNN consisting of two $5 \times 5$ convolutional layers with a stride of 1, each followed by a $2 \times 2$ pooling layer, and one fully connected hidden layer. The neurons in the pooling layer are simply LIF neurons, each of which connects to $2 \times 2$ neurons in the preceding convolutional layer with a fixed weight of $0.25$. Similar to [15, 32], we use elastic distortion [30] for data augmentation. As shown in Table 3, our proposed method achieves an accuracy of $99.49\%$, surpassing the best previously reported performance [32] with the same model complexity after 190 epochs.

**Fully Connected SNNs for EMNIST**  Table 4 shows that the HM2-BP outperforms the non-spiking ANN and the spike-based backpropagation (eRBP) rule reported in [21] significantly with less training epochs.

Table 2: Comparison of different models on N-MNIST

| Model | Hidden layers | Accuracy | Best | Epochs |
|---|---|---|---|---|
| Non-spiking CNN [23] | - | $95.02 \pm 0.30\%$ | - | - |
| Non-spiking CNN [22] | - | $98.30\%$ | $98.30\%$ | 15-20 |
| Non-spiking LSTM [23] | - | $96.93 \pm 0.12\%$ | - | - |
| Non-spiking Phased-LSTM [23] | - | $97.28 \pm 0.10\%$ | - | - |
| Spiking CNN (converted[*]) [22] | - | $95.72\%$ | $95.72\%$ | 15-20 |
| Spiking MLP [4] | 10000 | $92.87\%$ | $92.87\%$ | - |
| Spiking MLP [15] | 800 | $98.74\%$ | $98.74\%$ | 200 |
| Spiking MLP (STBP) [32] | 800 | $98.78\%$ | $98.78\%$ | 200 |
| Spiking MLP (this work) | 800 | $\mathbf{98.84 \pm 0.02\%}$ | $\mathbf{98.88\%}$ | $\mathbf{60}$ |

Only structures of SNNs are shown for clarity.[*] means the SNN model is converted from an ANN.

Table 3: Comparison of different spiking CNNs on MNIST

| Model | Network structure | Accuracy | Best |
|---|---|---|---|
| Spiking CNN (converted[a]) [6] | 12C5-P2-64C5-P2-10 | $99.12\%$ | $99.12\%$ |
| Spiking CNN (converted[b]) [7] | - | $92.70\%$[c] | $92.70\%$ |
| Spiking CNN (converted[a]) [27] | - | $99.44\%$ | $99.44\%$ |
| Spiking CNN [15] | 20C5-P2-50C5-P2-200-10 | $99.31\%$ | $99.31\%$ |
| Spiking CNN (STBP) [32] | 15C5-P2-40C5-P2-300-10 | $99.42\%$ | $99.42\%$ |
| Spiking CNN (this work[d]) | 15C5-P2-40C5-P2-300-10 | $99.32 \pm 0.05\%$ | $99.36\%$ |
| Spiking CNN (this work) | 15C5-P2-40C5-P2-300-10 | $\mathbf{99.42 \pm 0.11\%}$ | $\mathbf{99.49\%}$ |

[a] converted from a trained ANN. [b] converted from a trained probabilistic model with binary weights.
[c] performance of a single spiking CNN. 99.42% obtained for ensemble learning of 64 spiking CNNs.
[d] performance without data augmentation.

**Fully Connected SNNs for TI46 Speech** The HM2-BP produces excellent results on the 16-speaker spoken English letters of TI46 Speech corpus [16] as shown in Table 5. This is a challenging spatio-temporal speech recognition benchmark and no prior success based on SNNs was reported.

**In-depth Analysis of the MNIST and N-MNIST Results** Fig. 5(a) plots the HM2-BP convergence curves for the best settings of the first three experiments reported in the paper. The convergence is logged in the code. Data augmentation contributes to the fluctuation of convergence in the case of Spiking Convolution network. We conduct the experiment to see if our assumption used in approximating $\hat{\alpha}$ of (25) is valid. Fig. 5(b) shows that the value of $\hat{\alpha}$ of a randomly selected synapse does not change substantially over epochs during the training of a two-layer SNN (10 inputs and 1 output). At the high firing frequency limit, the S-PSP is proportional to $o_j^k \cdot o_i^k$, making the multiplicative dependency on the two firing rates a good choice in (25).

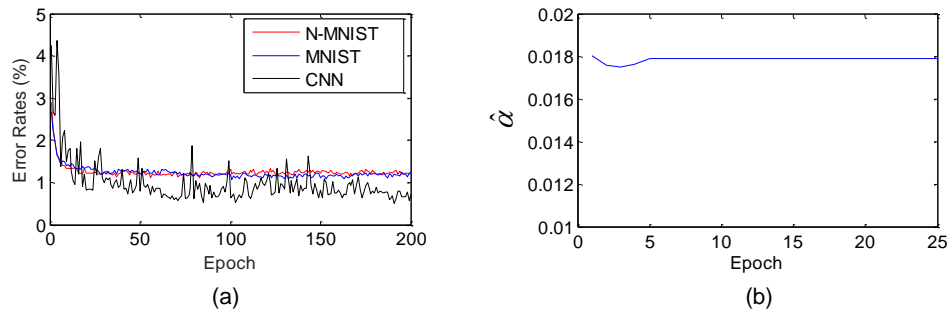

Figure 5: (a) HM2-BP convergence for the first three reported experiments; (b) $\hat{\alpha}$ v.s. epoch.

**Training Complexity Comparison and Implementation** Unlike [32], our hybrid method does not unwrap the gradient computation in the time domain, roughly making it $O(N_T)$ times more efficient than [32], where $N_T$ is the number of time points in each input example. The proposed

Table 4: Comparison of different models on EMNIST

| Model | Hidden Layers | Accuracy | Best | Epochs |
|---|---|---|---|---|
| ANN  [21] | 200-200 | 81.77% | 81.77% | 30 |
| Spiking MLP (eRBP) [21] | 200-200 | 78.17% | 78.17% | 30 |
| Spiking MLP (HM2-BP) | 200-200 | **84.31 ± 0.10%** | **84.43%** | **10** |
| Spiking MLP (HM2-BP) | 800 | **85.41 ± 0.09%** | **85.57%** | **19** |

Table 5: Performances of HM2-BP on TI46 (16-speaker speech)

| Hidden Layers | Accuracy | Best | Epochs |
|---|---|---|---|
| 800 | $89.36 \pm 0.30\%$ | 89.92% | 138 |
| 400-400 | $89.83 \pm 0.71\%$ | 90.60% | 163 |
| 800-800 | $90.50 \pm 0.45\%$ | 90.98% | 174 |

method can be easily implemented. We have made our CUDA implementation available online[1], the first publicly available high-speed GPU framework for direct training of deep SNNs.

## 4    Conclusion and Discussions

In this paper, we present a novel hybrid macro/micro level error backpropagation scheme to train deep SNNs directly based on spiking activities. The spiking timings are exactly captured in the spike-train level post-synaptic potentials (S-PSP) at the microscopic level. The rate-coded error is defined and efficiently computed and back-propagated across both the macroscopic and microscopic levels. We further propose a decoupled S-PSP model to assist gradient computation at the micro-level. In contrast to the previous methods, our hybrid approach directly computes the gradient of the rate-coded loss function with respect to tunable parameters. Using our efficient GPU implementation of the proposed method, we demonstrate the best performances for both fully connected and convolutional SNNs over the static MNIST, the dynamic N-MNIST and the more challenging EMNIST and 16-speaker spoken English letters of TI46 datasets, outperforming the best previously reported SNN training techniques. Furthermore, the proposed approach also achieves competitive performances better than those of the conventional deep learning models when dealing with asynchronous spiking streams.

The performances achieved by the proposed BP method may be attributed to the fact that it addresses key challenges of SNN training in terms of scalability, handling of temporal effects, and gradient computation of loss functions with inherent discontinuities. Coping with these difficulties through error backpropagation at both the macro and micro levels provides a unique perspective to training of SNNs. More specifically, orchestrating the information flow based on a combination of temporal effects and firing rate behaviors across the two levels in an interactive manner allows for the definition of the rate-coded loss function at the macro level, and backpropagation of errors from the macro level to the micro level, and back to the macro level. This paradigm provides a practical solution to the difficulties brought by discontinuities inherent in an SNN while capturing the micro-level timing information via S-PSP. As such, both rate and temporal information in the SNN is exploited during the training process, leading to the state-of-the-art performances. By releasing the GPU implementation code in the future, we expect this work would help move the community forward towards enabling high-performance spiking neural networks and neuromorphic computing.

**Acknowledgments**

This material is based upon work supported by the National Science Foundation under Grant No.CCF-1639995 and the Semiconductor Research Corporation (SRC) under Task 2692.001. The authors would like to thank High Performance Research Computing (HPRC) at Texas A&M University for providing computing support. Any opinions, findings, conclusions or recommendations expressed in this material are those of the authors and do not necessarily reflect the views of NSF, SRC, Texas A&M University, and their contractors.

## Footnotes

[1]`https://github.com/jinyyy666/mm-bp-snn`

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
