[Supplementary Material]

# Supplementary Material

## A    Gradient Computation for the Output Layer with Lateral Inhibition

Without loss of generality, it is assumed that lateral inhibition exists
between every pair of two output neurons. It is also assumed that the
weights for lateral inhibition are all fixed at the constant value of $w_0$.
As shown in Fig. A-6, the total post-synaptic potential (T-PSP) $a_i^m$ for
the neuron $i$ at the output layer $m$ can be unwrapped as

$$a_i^m = \sum_{j=1}^{r^{m-1}} w_{ij}\, e_{i|j}^m + \sum_{l \neq i}^{r^m} w_0\, e_{i|l}^m,$$

where the second term describing the lateral inhibition effects between
neurons in the same output layer.

The derivative of $a_i^m$ with respect to $w_{ij}$ is

$$\frac{\partial a_i^m}{\partial w_{ij}} = \frac{\partial}{\partial w_{ij}} \left( \sum_{j=1}^{r^{m-1}} w_{ij}\, e_{i|j}^m \right) + \frac{\partial}{\partial w_{ij}} \left( \sum_{l \neq i}^{r^m} w_0\, e_{i|l}^m \right)$$

$$= e_{i|j}^m \left( 1 + \frac{1}{\nu} \sum_{h=1}^{r^{m-1}} w_{ih} \frac{\partial e_{i|h}^m}{\partial o_i^m} \right) + \sum_{l \neq i}^{r^m} w_0 \frac{\partial e_{i|l}^m}{\partial o_l^m} g'(a_l^m) \frac{\partial a_l^m}{\partial w_{ij}}.$$

(A-26)

Figure A-6: The output layer with
lateral inhibition.

Similarly for the neuron $l$ of the output layer $m$, $a_l^m$ is given as

$$a_l^m = \sum_{p=1}^{r^{m-1}} w_{lp}\, e_{l|p}^m + \sum_{q \neq l}^{r^m} w_0\, e_{l|q}^m.$$

Therefore, $\frac{\partial a_l^m}{\partial w_{ij}}$ is

$$\frac{\partial a_l^m}{\partial w_{ij}} = w_0 \frac{\partial e_{l|i}^m}{\partial o_i^m} g'(a_i^m) \frac{\partial a_i^m}{\partial w_{ij}}.$$

(A-27)

Plugging (A-27) back into (A-26) and solving for $\frac{\partial a_i^m}{\partial w_{ij}}$ leads to

$$\frac{\partial a_i^m}{\partial w_{ij}} = \gamma\ e_{i|j}^m \left( 1 + \frac{1}{\nu} \sum_{h=1}^{r^{m-1}} w_{ih} \frac{\partial e_{i|h}^m}{\partial o_i^m} \right),$$

(A-28)

where

$$\gamma = \frac{1}{1 - \frac{w_0^2}{\nu^2} \sum_{l \neq i}^{r^m} \frac{\partial e_{i|l}^m}{\partial o_l^m} \frac{\partial e_{l|i}^m}{\partial o_i^m}}.$$

(A-28) is identical to (17) except that the factor $\gamma$ is introduced to capture the effect of lateral inhibition.

For the output layer with lateral inhibition, the term $\frac{\partial E}{\partial a_i^m}$ of the macro-level backpropagation defined in (14) is
the same as the one without lateral inhibition.