[Reviews · NeurIPS 2018]

Reviewer 1



In this paper, authors present a hybrid macro- and micro-level error backpropagation approach to train spiking neural networks. The idea is to depart from previous works by directly utilizing spiking activities and exact timings for error backpropagation - this enables measuring the direct impact of weight changes on the loss function and direct computation of gradients with respect to parameters. The error is defined, computed, and back-propagated with respect to (a): firing rates (macro-level) and (b): temporal sequences of action potentials for each neuron (spike trains) (micro-level). At micro-level, the model computes the spike-train level potential for each pre/post-synaptic spike train pair based on exact spike times. At macro-level, the model aggregates the effects of spike trains on each neuron's firing count. Authors propose a model to disentangle the effects of firing rates and spike-train timings to ease gradient computation at the micro-level. The model is compared against several baselines and outperforms the current state-of-the-art spiking neural network on two versions of MNIST dataset. Although spiking neural network is not my area of expertise, I have the following suggestions for authors: - lines 35-37: I think it is good to further explain how desired firing counts (labels) are computed for the output neurons and why it makes it challenging to compute gradient of the loss with respect to weights. - I think, in the contributions, it is informative to describe the temporal aspect and related challenges of the problem. For example, authors do not mention that the action potentials may decay over time. ---------------------------------------------------- Comment added after author response: I appreciate that the response sheds light on the characteristics of the proposed model; I hope authors report these analyses and comparisons in the final version of this paper. Also, the response and other reviews make me more confident about my assessment.

Reviewer 2



The paper presents a new approach for training spiking neural networks (SNNs) via backpropagation. The main contribution is a decomposition of the error gradient into a "macro" component that directly optimizes the rate-coded error, and a "micro" component that takes into account effects due to spike timing. The approach therefore creates a bridge between approaches that treat SNNs mainly like standard ANNs (optimizing the rate-coded error), and algorithms that work on the spike level such as SpikeProp, which have previously not been able to scale to larger problems. There are similarities to previously published methods such as from Lee et al. 2016 or Wu et al. 2017, which are discussed briefly, but there are novel contributions. Furthermore, the results on two tasks (MNIST and N-MNIST) show improvements over the state-of-the-art, although those improvements are small. Code for the algorithm is released in the supplementary material, which should allow reproduction of the results. I think the paper is a nice contribution to the field of SNN training, but neither very original nor a major breakthrough. Furthermore I think that the tasks are not chosen to allow highlighting the real strengths or limitations of the method, and that apart from the good empirical results there is no more insight why this method works better (e.g. on the MNIST task with Poisson rates, where spike timing should not matter much). This explains my vote for score 6 (barely above threshold). I am very familiar with the related literature, and I think the authors have cited the most relevant work, so I chose confidence score 5. The derivations are not too complex and seem to be correct and properly described. Regarding clarity, my main criticism are the figures, which are far from self-explanatory. One reason is the small size and the short captions, but also e.g. the arrows in Fig. 3 are barely visible. The paper is of relevance to the NIPS community, especially to the subcommunities interested in neuromorphic computing and computational neuroscience. I would however vote for a poster presentation, since these are not the biggest crowds at NIPS. The results advance the state of the art on relatively simple benchmark tasks by small margins. Detailed comments: The problem formulation starts from an error function which only makes sense for rate codes, so the approach seems to be limited to classical classification (or regression) tasks, but not for tasks where reproducing a spike pattern is the goal. The approach is thus not fundamentally more powerful than conversion methods, but achieves its advantage from treating timing effects at intermediate levels. It is interesting that this is an advantage even for tasks like static MNIST where timing plays almost no role, but no insight into how real spiking networks should learn or operate. In Section 2.2 the assumption is made that the total firing count can simply be calculated as the total post-synaptic potential divided by the threshold. This is only correct if there are no refractory periods or synchronous spikes that drive the potential way above threshold. In those cases, due to the reset of the membrane potential to zero, it seems that some of the potential is simply lost. In section 3 it is not precisely clear how the thresholds are set "depending on the layer". Also the conversion into Poisson spike trains does not describe the maximum input firing rate, which is known to be crucial for the success of SNN classifiers. A time window of 400 ms for the spike sequence is also pretty long, and it is not described how stimulus length influences performance. The simulation time step for N-MNIST is described as 0.55ms, which seems unusual. Please check if this is the correct value. The evaluation is not quite satisfactory, because the compared methods differ in architecture, training method, and training epochs, and no attempt is made to separate these hyperparameters. E.g. it would be interesting to see the performance of the macro/micro method with 200 epochs, to have a precise comparison with STBP. In general the improvements are small and no error bars are provided, which makes the comparison not quite conclusive. What the evaluation does not show is how much the micro vs. macro approach contribute to the success of the method. This would have required different tasks, where the timing of spikes (also on output level) plays a bigger role. Although the paper is generally well written, there are occasional grammar errors ("We particularly concern...") that should be checked and corrected. ------- Update after author response and discussion: While I appreciate the effort of the authors in running additional experiments during the rebuttal phase, I am not sure how much we should read into the comparison in 1.) in which the authors "quickly" implemented competing methods and show pretty terrible errors of the competing methods. It is of course impossible to judge whether the other methods were implemented correctly, but such a big difference is a big suspicious. Fig 1.b from author response reveals that the method does not improve after about 25 epochs and the error is consistently higher than for a CNN, which is a bit alarming as well. I also feel that my other detailed comments regarding the mathematical basis and details of the models were not at all addressed in the response. The figures would also definitely need some upgrade, but I understand that this cannot be handled in the feedback. Overall, I think this paper is still right around threshold, so I think it is justifiable to either accept or reject. I will keep my score at 6 and certainly not be upset with either decision.

Reviewer 3



[Response to rebuttal: I am very impressed and greatly appreciate the amount of experiments that the authors managed to successfully perform during the rebuttal phase. The additional results strengthen the empirical part of the paper significantly and further improve the experimental rigor of the paper. I am not entirely sure if I would rate the submission in the top 50%, but I would be somewhat upset if it got rejected. I therefore raise my overall score to an 8. I greatly appreciate the additional datasets and analysis in Fig. 1 of the rebuttal. I also want to encourage the authors to take some of the minor issues that were raised, for instance the quality and usefulness of the figures, into account, but I understand that one page of rebuttal is not sufficient to address all issues and I feel like the authors correctly prioritized addressing the issues raised.] The paper introduces a novel method for formalizing spiking neurons that allows for spiking neural networks which can be trained via stochastic gradient descent. In contrast to previous approaches to address this topic, the proposed model allows to incorporate temporal spiking dynamics while still maintaining differentiability w.r.t. the weights. This is achieved by formulating a rate-model (the macro-level) where the rate depends in a differentiable fashion on (approximate) leaky-integrate-and-fire dynamics (the micro level). In particular, the dependency of the rate on non-differentiable spikes is replaced by expressing the contribution of pre-synaptic spike trains on post-synaptic firing counts (and firing times) as a continuous function, termed “spike-train level post-synaptic potential” (S-PSP). This in turn allows for deriving closed-form gradient update equations for the weights. The method is evaluated on MNIST (fully connected network and convolutional neural network) and neuromorphic MNIST (fully connected network). On these experiments, the method outperforms competitor methods by a small margin. The widespread use of spiking neural networks in machine learning is arguably hindered by a lack of good training algorithms that scale well to large networks of neurons, particularly for the case of supervised learning. Due to the non-differentiable nature of spiking neurons, gradient descent cannot be applied in a straightforward way and several ways to alleviate this shortcoming have been proposed in the literature. The method presented in the paper addresses a timely and significant problem in an original and novel way that works well in the experiments shown. The writing is clear, with a well-written introduction and enough background for a machine learning practitioner to understand the method without having to resort to textbooks on spiking neuron models. I also greatly appreciate that the paper focuses clearly on machine learning applications and does not give in to the temptation of overloading the paper with additionally claiming biological plausibility (in a hand-wavy fashion). The authors also provide code to reproduce their results and also allowing for easy extension to experiment with additional ideas or set up additional experiments, which is definitely a plus. Besides minor issues, my main criticism is that the authors claim superior performance of the method even though they outperform competitors often by less than 0.2% accuracy on MNIST (comparing a single run with random initialization). While the claim of providing state-of-the-art results is technically not wrong, I would rather claim that the method performs on-par with competitor methods and further experiments (on more challenging datasets) are required for conclusive statements. However, I consider the latter overall a tolerable issue and suggest to accept the paper. My comments below are suggestions on how to further improve the quality and significance of the paper. (1 Decoupling of micro level) Could the authors clarify their particular choice of Eq. (25)? Are there any obvious alternatives (o_j and o_k have a linear effect, but does it necessarily have to be multiplicative or is this a choice motivated by mathematical convenience)? Can the authors provide some quantitative analysis on how sensitive the approximation is w.r.t. the assumption that \hat{\alpha} does not change substantially (e.g. a histogram of how much \hat{\alpha} actually changes in typical experiments, or some comparison of numerical gradients vs. gradients based on the approximation)? (2 Error bars) Even though it has become widely accepted in the neural network community to report results on single runs, without sensitivity-analysis w.r.t. hyper-parameter settings, I want to actively encourage the community to re-introduce more experimental rigor by setting high standards. Whenever possible “with reasonable effort” it would be great if results were reported with “error bars” (e.g. mean/median accuracy over a small number of runs with an appropriate error-bar measure). While I do not strictly require the authors to re-run multiple instances of their experiments for this publication, I would consider a higher score if error bars were provided. (3 Additional datasets) MNIST (and neuromorphic MNIST) seems to have saturated as a data-set to compare different methods for spiking neural network training. It would be an interesting move to start evaluating on more challenging datasets. FashionMNIST should be really straightforward, but it would also be interesting to see more challenging problems on data-sets with temporal / event-driven signals in the long run. (4 Figures) The clarity of figures, particularly of figure captions could still be improved. Given the limited space of the manuscript you might want to push some parts of the derivation or some details of the experiments to the appendix. -) Fig. 1: there are lots of different kinds of arrows in the figure, a bit more space and text in the figure as well as the caption would help. Also, what does the “(+/-) Multiple Spikes” refer to? -) Fig. 2: What does the shaded yellow area correspond to? Why is there no re-set of the middle curve after the second post-synaptic spike? -) Fig. 4: The figure does not add much more clarity compared to the equations, but I think with a bit of polishing (and more space) it could add value to the paper. (5 Control exp.) For the spiking conv net on MNIST, the results are better than for the MLP (Table 3 vs. 1) which is not too unexpected. However, for Table 3 data augmentation was used as well, but not for Table 1. It would be interesting to see how the result in Table 3 changes without data augmentation – i.e. how much of a performance improvement is gained by using a conv net over an MLP? (6 Convergence speed) One advantage of the proposed method is that it seems to converge faster to high accuracies compared to competitor methods. Since the final accuracies are almost on-par, it would be interesting to have more quantitative analysis on convergence behavior (i.e. convergence curves for the different methods). I understand, of course, that this might not be easily possible since the competitor papers might not report such data and would require re-implementations. If such curves cannot be easily obtained, I do not expect the authors to re-implement other methods during rebuttal, but if possible it would be nice to include convergence curves (i.e. accuracy vs. training epochs) for the proposed method, such that future methods can compare against these results. (7 Supplementary code) The accompanying release of code in the style of an easily extendable and customizable framework is a big plus and could potentially be very beneficial to the community. For the final release of the code please consider adding a bit more documentation and some documented examples. I also highly appreciate that the authors tested their code on Windows and Linux.